# High syphilis prevalence and incidence in people living with HIV and Preexposure Prophylaxis users: A retrospective review in the French Dat'AIDS cohort

Thomas Lemmet[1]*, Laurent Cotte[2,3], Clotilde Allavena[4], Thomas Huleux[5], Claudine Duvivier[6,7,8,9], Hélène Laroche[10], André Cabie[11], Pascal Pugliese[12], Thomas Jovelin[4], Marine Maurel[13], Cyrille Delpierre[13], David Rey[1]

1 Le Trait d'Union, HIV Infection Care Center, Strasbourg University Hospital, Strasbourg, France, 2 Department of Infectious Diseases, Croix-Rousse Hospital, Hospices Civils de Lyon, Lyon, France, 3 INSERM U1052, Lyon, France, 4 Infectious Diseases Department, Nantes University Hospital, Nantes, France, 5 Service Universitaire des Maladies Infectieuses et du Voyageur, Centre Hospitalier DRON, Tourcoing, France, 6 AP-HP - Necker-Enfants Malades Hospital, Infectious Diseases Department, Necker-Pasteur Infectiology Center, Paris, France, 7 IHU Imagine, Paris, France, 8 Institut Cochin - CNRS 8104 - INSERM U1016 - RIL (Retrovirus, Infection and Latency) Team, University of Paris, Paris, France, 9 Institut Pasteur, Institut Pasteur Medical Center, Paris, France, 10 Immuno-Hematology Clinic, Assistance Publique, Hôpitaux de Marseille, Hôpital Sainte-Marguerite, Marseille, France, 11 Infectious and Tropical Medicine Unit, University Hospital of Martinique, Fort-de-France, France, 12 Université Côte d'Azur, CHU de Nice, Nice, France, 13 UMR1027, INSERM, Université Toulouse III Paul-Sabatier, Toulouse, France

* thomas.lemmet@chru-strasbourg.fr

**Data Availability Statement:** All relevant datasets for People living with HIV (PLWH) and PrEP users whose data had been analyzed in our study are

## Abstract

### Background

In the past years, we observed a sharp increase of Syphilis, especially among male who have sex with male (MSM), either HIV-infected, or on pre-exposure prophylaxis (PrEP). Our aim was to assess syphilis prevalence and incidence among people living with HIV (PLWH) and PrEP users.

### Methods

PLWH were included from 2010 to 2020 and PrEP users from 2016 to 2020 from the Dat'AIDS French cohort. We calculated syphilis prevalence and incidences for first infections, re-infections, and iterative infections (> 2 times). T-Tests, Wilcoxon tests and Chi2 test were used for descriptive analysis and multivariate logistic regression models were used to estimate Odds ratios (OR) and 95% confidence intervals (95% CI) for factors associated with syphilis.

### Results

Among the 8 583 PLWH, prevalence of subject with past or present syphilis was 19.9%. These subjects were more likely MSM or transgender and aged over 35 years, but prevalence was lower in AIDS subjects. Same pattern was seen for incident infection and re-infection. Incidence was 3.8 per 100 person-years for infection and 6.5 per 100 person-years for

within the paper and its Supporting information files. These data can be used by interested researchers to replicate our findings in their entirety.

**Funding:** The authors received no specific funding for this work.

**Competing interests:** D. Rey declared having received grants from ViiV Healthcare (expert board participation and fee for lecture) and Mylan (fee for lecture). C. Allavena declared having received grants from Gilead, Janssen, MSD, ViiV Healthcare. L. Cotte declared having received grants from Abbvie, Gilead Science, Janssen Cilag, MSD, ViiV Healthcare. For the remaining authors no other conflicts of interest were declared.

re-infection. Among 1 680 PrEP users, syphilis prevalence was 25.8%, with an estimated 7.2% frequency of active syphilis. Risk of syphilis infection was higher in male and increased with age. Incidence was 11.2 per 100 person-years for infection and 11.1 per 100 person-years for re-infection.

## Conclusion

Syphilis prevalence and incidence were high, especially in older MSM with controlled HIV infection and PrEP users, enhancing the need to improve syphilis screening and behavioral risk reduction counseling among high-risk subjects.

## Introduction

Syphilis is a sexually transmitted infection (STI) caused by the bacteria *Treponema pallidum*. It is a worldwide spread infection, highly prevalent among male who have sex with male (MSM), and infects an average of 11.8% of MSM in reporting countries in 2019 according to the World Health Organization. Untreated, syphilis can cause serious complications, and increases the risk of human immunodeficiency virus (HIV) acquisition [1–4]. The association between HIV and syphilis can be explained by behavioral factors but also because of pathological mechanisms, such as facilitated HIV acquisition through ulcer and genital inflammation caused by syphilis [5–7].

In 2018 in France, 1762 new diagnoses of early syphilis have been notified, of which 79% occurred in MSM. Although the number of reported cases seems to stabilize since 2016, it remains higher than in 2010, and 32% of the MSM were co-infected with HIV according to "Santé Publique France" [8]. Due to biologic and epidemiologic associations between syphilis and HIV infection [9], this rate remains concerning.

Pre-Exposure Prophylaxis (PrEP), introduced in France in 2016 following demonstration of its effectiveness in reducing HIV acquisition in high-risk populations [10, 11], has been increasingly prescribed. As a result, increase in unsafe sex [12] has been described. Some studies have shown that PrEP users are more at risk of bacterial STIs [13, 14], including syphilis, and it has been suggested that syphilis was associated with HIV incidence among MSM taking PrEP [15]. Therefore, as for HIV, monitoring populations at risk for syphilis is becoming more relevant for targeting prevention and treatment to decrease syphilis transmission.

The objectives of this study are to evaluate the prevalence of syphilis, whether earlier or older, in a cohort including people living with HIV (PLWH) and subjects taking PrEP in France, and to explore the incidence of new infection and subsequent re-infection of syphilis during their medical follow-up, between 2010 and 2020 for PLWH, and from 2016 for PrEP users.

## Materials and methods

### Study design

This multicenter retrospective analysis was performed using longitudinal data from the French Dat'AIDS cohort (NCT 02898987 ClinicalTrials.gov) [16]. In 2010, Dat'AIDS represented a collaboration between 17 major French HIV clinical centers that used a common electronic medical record system (NADIS® software) for the follow-up of PLWH, hepatitis B virus

(HBV) or hepatitis C virus (HCV). Concerning PrEP users, data were collected through a specific module of the NADIS® program dedicated to sexual health implemented in 2016.

Data collection was approved by the French National Commission on Informatics and Liberty (CNIL 2001/762876; MR004 2210731v.0). All subjects signed an informed consent form before being included in this database. Patient-related data obtained during medical encounters are recorded in a structured database. Data quality is ensured by automated checks during data capture, regular controls, annual assessments, and ad hoc processes before any scientific analysis is performed.

French guidelines recommend testing PLWH for STIs, including syphilis, at initial medical examination, then at least once a year or according to clinical signs and/or risky sexual behaviors. For PrEP subjects, it is recommended to screen STIs including syphilis at initial visit, 1 month after PrEP initiation, and then every 3 months.

The aim of this study is to estimate prevalence and incidence of syphilis among a cohort of subjects either living with HIV or under PrEP treatment.

## Study population

For this study, we selected PLWH followed-up between January 1st 2010 and December 31st 2019 in the Dat'AIDS database, with a treponemal test available within 30 days after initial medical assessment. Different treponemal tests were used: qualitative TPHA (Treponema pallidum Haemagglutination) or FTA (Fluorescent Treponemal Assay) or ELISA (enzyme immunoassay) or CMIA (Chemiluminescence Microparticle Enzyme Immunoassay). Non-treponemal tests were also used: either VDRL (Venereal Disease Research Laboratory) or RPR (Rapid Plasma Reagin), which both give quantitative results when positive. In France, before 2018, a serological syphilis evaluation always included a combination of treponemal and non-treponemal tests, while from June 2018, a treponemal test was first recommended, and only when positive, completed in a second step by a non-treponemal test.

Due to the guidelines change in 2018, we did not dispose of treponemal and non-treponemal tests for all of the subject at baseline. Thus, we assessed the prevalence of positive treponemal tests at baseline, which gather subjects with either past or early syphilis, whereas patients with a negative treponemal test at baseline included all subjects without syphilis history nor current disease. We were not able to formally differentiate past and early syphilis by using non-treponemal test, as it is difficult to define a titer cut-off for this purpose. Nevertheless, we tried this analysis with a 1/8 cut-off.

## Measures

Treponemal tests were considered as positive when qualitative TPHA, FTA, ELISA or CMIA were positive.

Collected data consisted in demographic characteristics including age and gender for both group of subjects. For PLWH, HIV risk factor, 1993 CDC classification, HIV viral load and CD4 cell count at inclusion were collected at the same time as the syphilis serology.

**Incident first infection.** It corresponds to the first positive treponemal test during follow-up, when negative at inclusion. Follow-up duration is defined by the time elapsed between initial assessment date at inclusion, and either the date of first positive treponemal test (first infection), or the date of the last available negative syphilis test (non-infection).

**Incident re-infection.** For all infected subjects, cases of incident re-infection were defined by positivation of a previously negative non-treponemal test (VDRL or RPR) or by a 2 dilutions increase in quantitative VDRL after at least 90 days following first positive treponemal test. Cut off was set at 90 days as it is the recommended delay for re-assessing VDRL after

treatment of a primary syphilis [17]. Of note, we assumed that any VDRL increase at 90 days or more is a re-infection; therefore, we omit the possibility of an undiagnosed and untreated syphilis.

Follow-up duration is defined by the time elapsed between initial assessment date, and either the date of first VDRL positivation or of 2 dilutions increase in quantitative VDRL, or the date of last available negative test (absence of re-infection).

**Iterative infections (> 2 times).**   Iterative infections were defined as any re-infections which occurred more than twice in the same subject, according to the criteria of re-infection: negative VDRL test which becomes positive, or a 2 dilution increase of quantitative VDRL, in subjects with positive treponemal test.

## Statistical analysis

The associations between covariates and the outcome were teste in bivariate analyses using T-Tests or Wilcoxon tests (when normality assumption was not respected) for continuous variables, and Chi2 tests for categorical variables. Continuous variables are presented as means and standard deviations (SD), or if non-normal as medians and interquartile ranges (IQR). Categorical variables are presented as numbers with percentages for each category.

We used a multivariate logistic regression model to investigate the probability to be infected or re-infected adjusted on covariates in HIV subjects and in PrEP users separately. Considering the incompleteness of the chronological data, we decided not to proceed to a survival analysis.

The model run for HIV subjects was adjusted on age, sex, sex of sexual partners, CDC Stage, HIV viral load, CD4 level, CD8 level and year of syphilis diagnosis.

For PrEP users, the model conducted was adjusted on age, sex, and year of syphilis diagnosis.

These variables were selected from the literature as factors potentially associated with the risk of syphilis infection. The underlying assumptions of the multivariate logistic regression were tested and the residuals analysed. No violations were found and the goodness of fit of the model was good.

All analyses were run using R v.3.1.2 along with its integrated development environment, RStudio.

## Results

### Participant characteristics at baseline

Between January 1$^{st}$ 2010 and December 31$^{st}$ 2019, 9 040 subjects living with HIV and 1 680 PrEP users were included with valid treponemal test, among 71 214 PLWH, and 7 445 PrEP users constituting the Dat'AIDS cohort. PLWH with baseline CD4 and HIV viral load missing data (n = 429) were excluded, as well as 10 and 18 PLWH, because of discordant serology within 30 days after baseline assessment and file inconsistency, respectively. The final study population (Fig 1) included 8 585 subjects living with HIV and 1 680 PEP users. Participant characteristics are shown in Table 1.

### Baseline syphilis prevalence

At baseline, we reported 1 708 PLWH (19.9%) and 434 PrEP users (25.8%) with a positive treponemal test (Table 1).

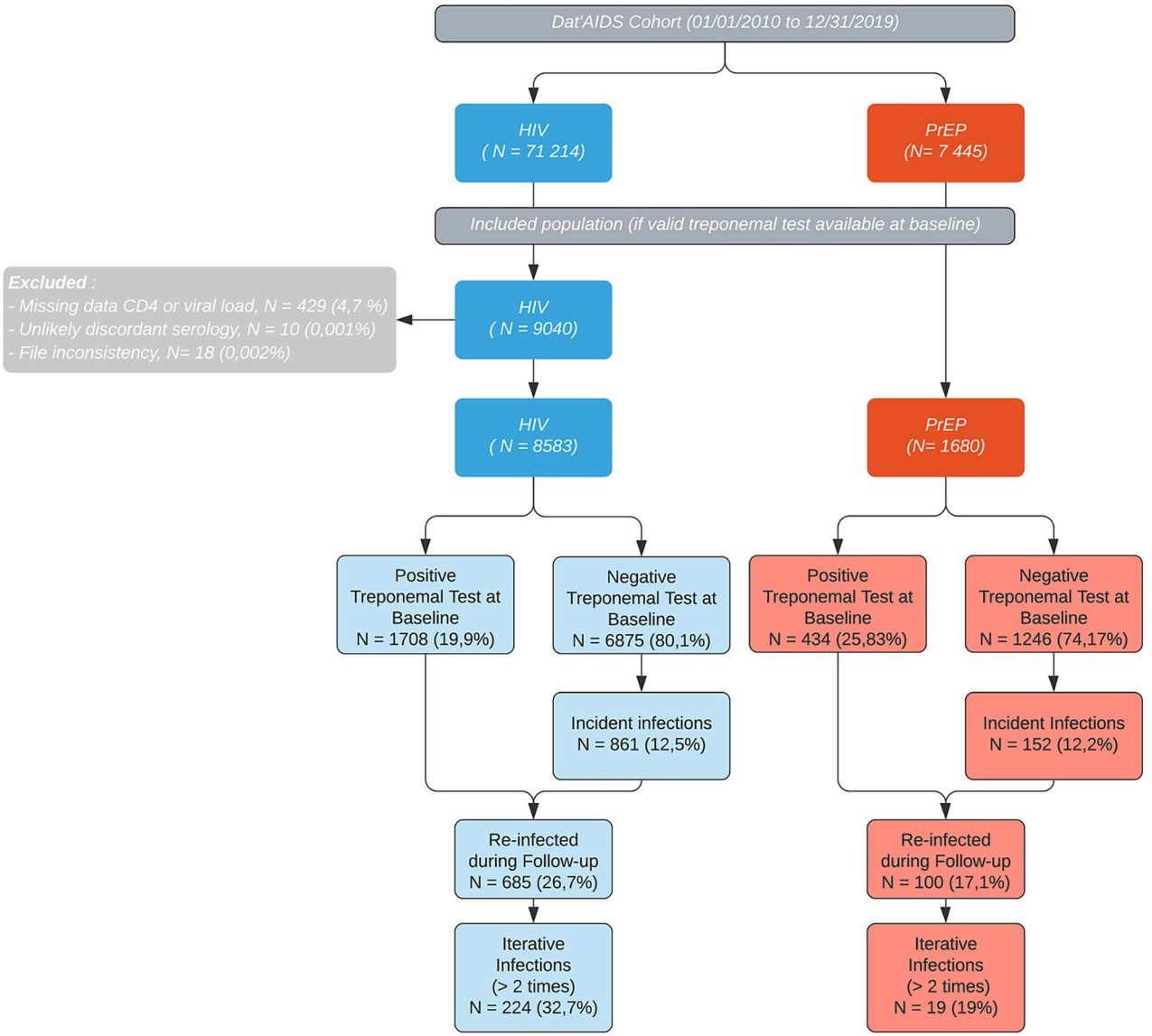

**Fig 1. Flowchart.**

PLWH with syphilis were mostly male (92.4% vs. 69.2%, p<0.001) and MSM (77.6% vs.44%, p<0.001). There was no significant difference in syphilis infection rate according to HIV viral load at baseline.

In multivariable model, female had a lower risk of being infected by syphilis at baseline than male (OR (95% CI) = 0.31(0.24; 0.39)). Transgenders had a higher risk compared to male (OR (95% CI) = 1.81 (1.14; 2.86)). Moreover, the risk of syphilis increased with age (higher risk in 35–47 years old subjects and 47–88 years old subjects compared to 15–35 ones (OR (95% CI) respectively at 1.34 (1.16; 1.54) and 1,37 (1.18; 1.58)), was higher in MSM compared to heterosexual subjects and those with other HIV risk factors (OR (CI 95%) = 3,05 (2.60; 3.58) and 1.44 (1.10; 1.86) respectively), and lower in CDC Stage C subjects (OR (95% CI) = 0.80 (0.66; 0.96)).

**Table 1. Characteristics of participants at enrollment.**

| Characteristics | HIV No. (%) | PrEP No. (%) |
|---|---|---|
| **Total** | 8585 | 1680 |
| **Syphilis** | | |
| Negative, n(%) | 6875 (80.1) | 1246 (74.17) |
| Positive, n(%) | 1708 (19.9) | 434 (25.83) |
| **Type treponemal test** | | |
| TPHA qualitative, n(%) | 7812 (91.02) | 1140 (67.86) |
| FTA, n(%) | 60 (0.7) | 0 (0) |
| CMIA Qualitative, n(%) | 342 (3.98) | 25 (1.49) |
| ELISA, n(%) | 369 (4.3) | 515 (30.65) |
| **Gender** | | |
| Male, n(%) | 6332 (73.77) | 1621 (96.49) |
| Female, n(%) | 2173 (25.32) | 20 (1.19) |
| Transgender Man to Woman, n(%) | 78 (0.91) | 38 (2.26) |
| Transgender Woman to Man, n(%) | 0 (0) | 1 (0.06) |
| **Age** | | |
| Mean (SD) | 42.5 (12.8) | 38.5 (10.7) |
| Median (IQR) | 41 (33–51) | 37 (30–47) |
| **Sex of sexual partners** | | |
| MSM, n(%) | 4350 (50.68) | |
| Heterosexual, n(%) | 3589 (41.82) | |
| Unknown, n(%) | 644 (7.5) | |
| **AIDS (CDC Stage C)** | 1213 (14.13) | |
| **CD4 count (baseline)** | | |
| Mean (SD) | 691 (352) | |
| Median (IQR) | 650 (462–879) | |
| $\geq$ 500, n(%) | 6032 (70.28) | |
| [200;500], n(%) | 2148 (25.03) | |
| < 200, n(%) | 403 (4.70) | |
| **HIV Viral load baseline** $\leq$ 50, n(%) | 7614 (88.71) | |

TPHA: Treponema pallidum Haemagglutination; FTA: Fluorescent Treponemal Assay; CMIA: Chemiluminescence Microparticle Enzyme Immunoassay; ELISA: enzyme immunoassay; MSM: Male who have sex with male; AIDS: acquired immune deficiency syndrome.

For PrEP users, 434 were infected at baseline (25.8%). After adjustment, female were at lower risk of being infected by syphilis than male (OR (95% CI) = 0.17(0.01; 0.82)), and transgenders (only male to female were involved) had a higher risk (OR (95% CI) = 3.40 (1.75; 6.59). As for PLWH subjects, risk of infection increased with age: subjects aged 32–43 and 43–78 had a higher risk compared to subjects aged 18–32 (OR (95% CI) = 1.49 (1.12; 1.99) and 1.54 (1.16; 2.04) respectively).

Adjusted odds ratios for factors associated with syphilis infection at baseline are shown in Fig 2.

## Estimation of early syphilis according to quantitative VDRL

Concomitant quantitative VDRL was available for 56% of PLWH with a positive treponemal test, of whom 54.1% had a VDRL $\geq$ 1/8. When extrapolating this data to our whole population, we can estimate the prevalence of PLWH with a VDRL $\geq$1/8 at baseline to be 10.8%.

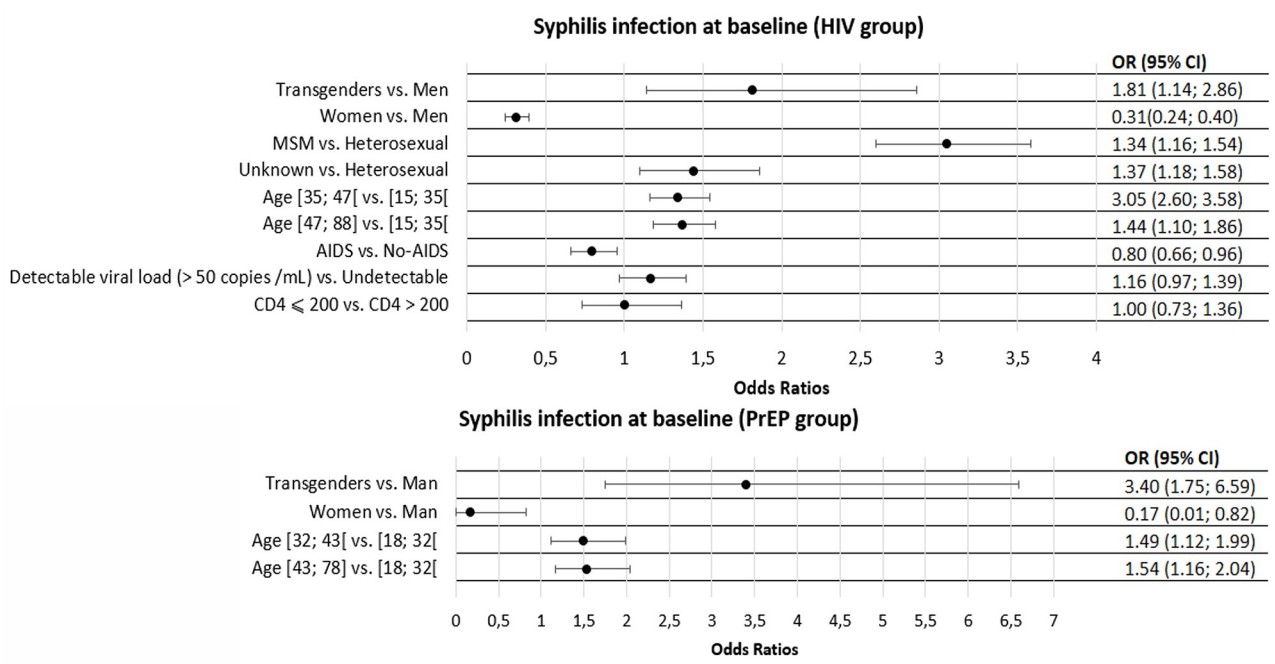

**Fig 2. Adjusted odds ratios for factors associated with syphilis infection at baseline among participants.**

Similarly, concomitant quantitative VDRL was available for 54.4% of PrEP users with syphilis at baseline, of whom 28.0% had a VDRL ≥ 1/8., giving an estimation of 7.2% of PrEP users with a VDRL ≥1/8.

## Incident first infection

Among 8121 subjects free of syphilis at baseline screening, i.e. 6875 PLWH and 1246 PrEP users, 1013 (12.5%) were infected during follow-up, including 861 PLWH (12.5%) and 152 subjects under PrEP (12.2%). These infections occurred between 33 days and 21 years after first treponemal test, with a median of 2 years and 5 months.

Men represented 95% of infections in PLWH, of whom 86% were MSM. Infected and non-infected subjects had the same age (mean: 42.4 vs 42 years, p = 0.32). Among infected subjects, AIDS was less frequent (9.8% vs 15.6% p<0.001) and CD4 cell count was higher (mean of 779 vs 670 p<0.001).

In multivariable model, we observed the same conclusions as at baseline (Fig 3). Female were at lower risk than male to be infected by syphilis (OR (95% CI) = 0.28 (0.18; 0.43)), while there was no difference between male and transgenders (OR (95% CI) = 1,04 (0.49; 2.06)). Older subjects had a greater risk of being infected (between 35–47 years old and 15–35 years old, OR (95% CI) = 1.81 (1.49; 2.21)). MSM and subjects with other HIV risks factors had a significantly higher risk compared to heterosexuals (OR (CI 95%) = 6.60 (5.10; 8.66); and 1.65 (1.05; 2.53) respectively). Finally, CDC stage 3 subjects had a lower risk (OR (95% CI) = 0.99 (0.75; 1.30)) as well as subjects with a detectable viral load at baseline (OR (95% CI) = 0.63 (0.45; 0.86)).

Concerning subjects on PrEP, a total of 152 were diagnosed with syphilis during follow up (Fig 3). They were older than non-infected subjects, with a median age of 38.5 years (vs 36) but the difference was not significant. The studied population did not include any female and only

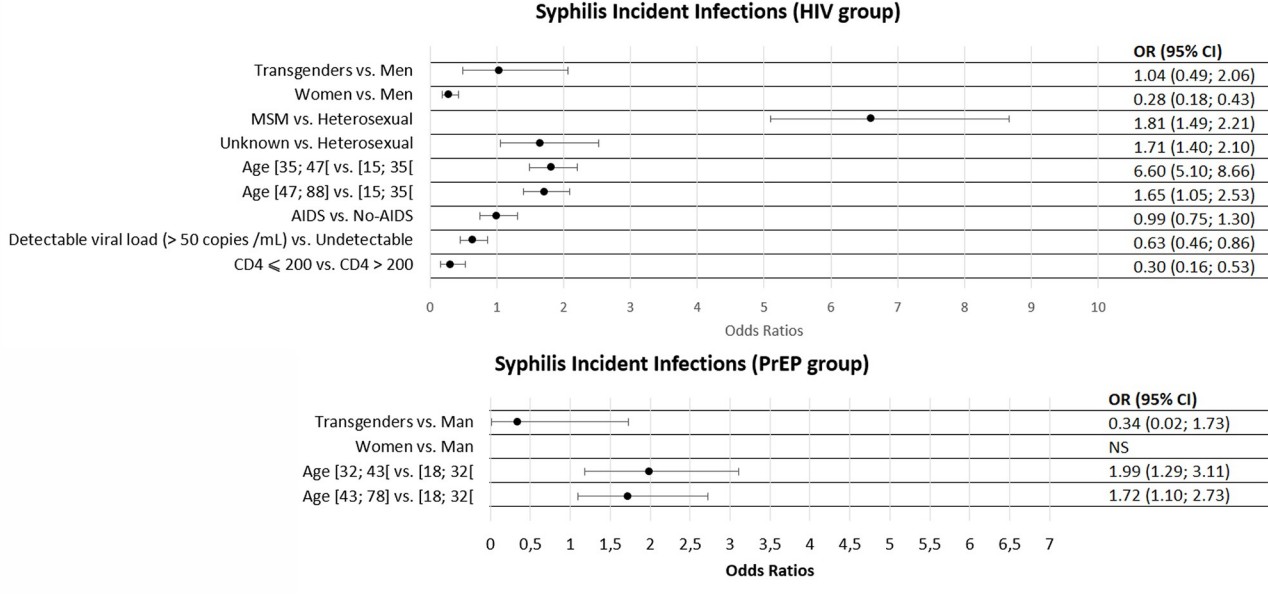

**Fig 3. Adjusted odds ratios for factors associated with syphilis infection during follow up among participants.**

one transgender (male to female). As for PLWH, age was also associated to the risk of syphilis infection, (higher risk in 32–43 years old compared to 18–32 years old (OR (95% CI) = 1.99 (1.29; 3.11)).

## Incident re-infections

During follow up, 785 (24.9%) subjects were reinfected at least once among those with an history of syphilis (i.e. either with a positive treponemal test at baseline, or with incident first infection), including 685 PLWH and 100 on PrEP, resulting in a re-infection rate of 26.7% and 17.1% respectively.

Among 685 reinfected PLWH, most of them were MSM (90.4%), aged 35–47 (40.6%) or older than 47 years (34.3%), with an earlier clinical stage of HIV infection (86.4% CDC A) and a higher CD4 cell count (median 749 vs 686).

After adjustment (Fig 4), female had a lower risk of being reinfected by syphilis than male (OR (95% CI) = 0.51 (0.23; 1.02)). The risk of re-infection increased with age, as subjects aged 35–47 had a higher risk than those aged 15–35 (OR (95% CI) = 1.31 (1.04; 1.67)) and was higher among MSM (OR (CI 95%) = 2.71 (1.89; 3.96). In addition, having a detectable viral load or a CD4 cells count below 200/μL at baseline were associated with a lower risk of re-infection (OR (95% CI) = 0.44 (0.29; 0.65) and OR (95% CI) = 0.35 (0.19; 0.84) respectively).

Concerning PrEP subjects, risk of re-infection did not differ according to age. Sex was not studied as there was only one female in this population.

## Iterative infections (> 2 times)

Among 685 reinfected PLWH, 32.7% (n = 224) were reinfected more than once (2 to 6 times). Among 100 reinfected PrEP subjects, 19% (n = 19) were reinfected more than once (2 to 3 times).

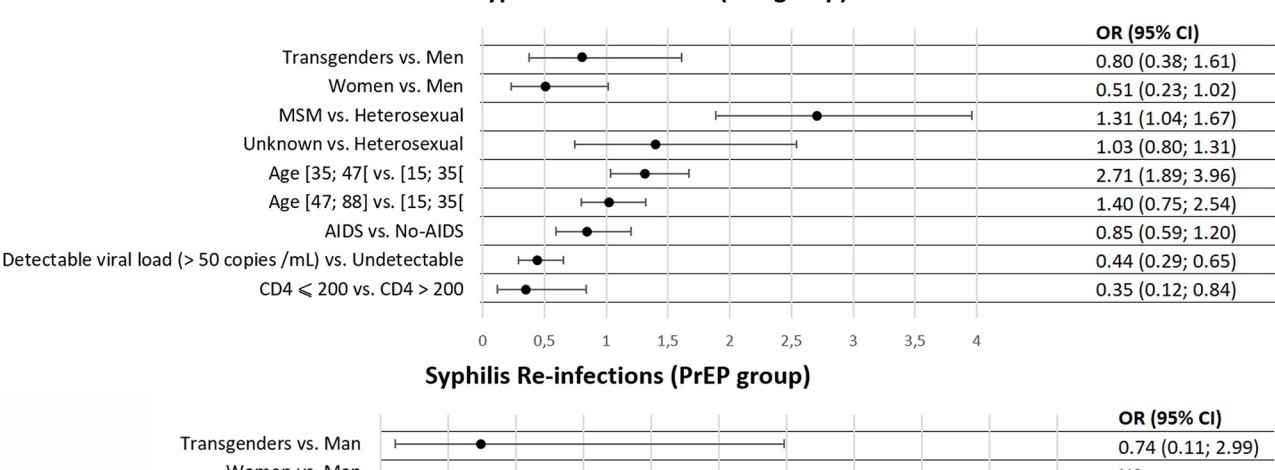

**Fig 4. Adjusted odds ratios for factors associated with syphilis re-infection during follow-up among participants.**

## Incidence rate

Incidence rate of first syphilis infection among PLWH was 3.75 cases per 100 person-years, with a total person-time of observation (follow up time) of 22834 years, and for PrEP users was 11.22 cases per 100 person-years, with a total person-time of observation of 1355 years. Incidence rate of syphilis re-infection among PLWH was 6.48 cases per 100 person-years, with a total person-time of observation (follow up time) of 10566 years, while it was 11.08 cases per 100 person-years for PrEP users, with a total person-time of observation (follow up time) of 902 years.

Of note, there were no available follow up data for 1781 PLWH (20.8%) and for 339 PrEP users (20.2%), for whom we disposed only of their baseline data. Thus, they were not considered in the analysis.

## Discussion

In this study, we found a high prevalence of syphilis among PLWH (19.9%) at initial examination, but this result did not differentiate early syphilis from a persistent positive antibody test after treatment; therefore, comparisons with other studies are somewhat difficult. Nevertheless, and according to our estimation using quantitative VDRL data, 10.8% of PLWH presented with an active syphilis at initial examination. It is consistent with latest data of the literature: according to CDC report of 2018, 7% of HIV-positive MSM attending selected STI clinics were diagnosed with primary or secondary syphilis [18]. A recent study in Turkey from 2010 to 2018 found a 8% prevalence of syphilis and HIV co-infection [19], while more recently, a similar study in Thailand found an even higher prevalence of syphilis (14.3%) in acute HIV-infected MSM [20].

Risk factors associated with initial syphilis among PLWH were male sex or transgender, age older than 35 years, MSM, and controlled HIV infection on ART (antiretroviral therapy). The

same pattern was seen for incident infection during follow up, and it was even more obvious regarding re-infection. Thus, non-controlled HIV infection at baseline was associated with a lower risk of syphilis infection or re-infection, as shown by a lower re-infection rate in subjects at CDC C stage, or with a detectable viral load, or with a CD4 cell count below 200/ μL. Perception of HIV transmission risk while HIV was not controlled might explain behavior adaptation, resulting to lower syphilis infections. Concerning older age as a risk factor for a positive syphilis serology at inclusion, our hypothesis is that the risk would rather be linked to a repeated exposition to syphilis over time, which increases the risk of infection. In this cohort, re-infection rate was alarming among PLWH, as it concerned more than one on four subjects coinfected at baseline (26.7%). Moreover, it seemed that some subjects are particularly affected, as 32.9% of them were reinfected more than once. Incidence rate was 3.77 cases per 100 person-years for infection, and 6.47 cases per 100 person-years for re-infection, confirming that syphilis occurs more frequently in previously infected subjects, and reflecting the lack of prevention measures. As a comparison, a recent study found a comparable incidence rate of syphilis of 4,7 per 100 person-years in a cohort of PLWH from the United States [21]. However, unlike ours, this study found higher rates among younger MSM and those with detectable HIV viral load. Another study from Thailand [20] found an incidence of syphilis of 10.2 per 100 person-years, with a re-infection rate of 41.4%. Some studies have shown that syphilis might increase HIV viral load and decrease CD4 cell count in PLWH [22, 23], increasing risk of HIV transmission. However, another recent study showed that it does not increase the risk of HIV viral load elevation in PLWH under effective treatment [24]. Nowadays, it is recommended to screen PLWH at least annually for syphilis. Even though our study is not designed to verify the link between syphilis, HIV viral load and CD4 cell count, our data support the idea of a strict monitoring among high-risk subjects, as those with a known history of syphilis.

Concerning PrEP users, 25.8% had a positive syphilis serology at baseline, and we estimated that 7.2% had an early syphilis at PrEP initiation. This rate is close to those previously reported, as other studies found a prevalence of syphilis at PrEP initiation of 7.8% to 13.3% [12, 14, 15, 25]. Syphilis incidence rate for first infections and re-infection were similar to other studies, as they described an incidence of syphilis among PrEP users from 7.3 to 14.7 cases per 100 person-years. Risk compensation has been suggested in PrEP users to explain these high incidences of syphilis and other non HIV STIs, by not perceiving an increased risk [26]. It has also been showed a higher incidence in STIs, including syphilis, in consistent PreP users compared to those who discontinued the prophylaxis [27]. These results among PrEP users reinforce once again the necessity of a timely and efficient screening and treatment of syphilis, and adherence to medical follow-up.

When comparing PLWH and PrEP users, syphilis incidence appears to be significatively higher for infection and re-infection among PrEP users. This might be explained by the more often risky sexual behaviour [12] observed in the latter. One hypothesis is that PrEP might bring users into contact with others individuals engaged in social and sexual network where rates of STIs including syphilis may be higher [28]. Besides, frequency of STI monitoring in PrEP users (every 3 months) compared to PLWH (at least once a year), could explain an artificial increase of incidence rate in PrEP users. Nevertheless, reinforcing screening policy, associated with treatment of cases, might at last lead to a decrease of syphilis incidence among PrEP users. This hypothesis was already suggested concerning gonorrhea, as shown by a mathematical model [29].

One limitation of our study is that our analysis did not differentiate active syphilis from persistent positive antibody test after treatment, nor syphilis stage of diseases such as primary or secondary. Indeed, it was hardly possible with isolated serologic results (i.e; without kinetic) to make this distinction and gathering clinical data or history of serologic tests for all subjects

was unachievable in this retrospective cohort. Thus, prevalence in our study is higher than previously observed in literature, as we included any positive treponemal test at baseline in our analysis. Therefore, we decided to estimate the proportion of early syphilis using an ad hoc cut off for quantitative VDRL ($\geq$ 1/8). Although seemingly relevant, this choice is arbitrary and we do not have data using a similar definition. A second limitation is the large number of missing data regarding treponemal test results among the whole Dat'AIDS cohort. Indeed, we were able to include only 12,7% of the PLWH and 22,6% of the PrEP users, despite the recommendation of systematic syphilis screening for all PLWHIV and PrEP users. This might be due to syphilis screening at a different time, or somewhere else, or to the lack of data entry. Finally, syphilis follow-up was not standardized, nor were serologic tests, which could result in misdiagnosis in such a retrospective analysis.

Even though we decided to estimate incidences, the model used was not optimized to do so. Given the uncertainty about the dates entered in the database, we have chosen to use a logistic regression instead of a survival analysis. This model strengthens our confidence in the analysis of risk factors for infection and re-infection, but since follow-up time is not taken into account, incidence is only estimate to take with caution.

Finally, results of our study might not be generalizable to all populations, as some populations were less represented such as female or transgenders among PrEP users (only 20 female and 39 transgenders included). However, this is consistent with demographics of PrEP users. In conclusion, this study underscores that syphilis prevalence and incidence are high in PLWH, specifically in older MSM with controlled HIV infection by ART, as well as in PrEP users in France. Even though current syphilis epidemic among MSM seems to have stabilized the past few years, our data highlight the necessity to improve syphilis screening frequency among PrEP users and PLWH at high risk, especially those with known history of syphilis, and to reinforce behavioral risks reduction counseling.

## Supporting information

**S1 Dataset.**
(XLSX)

**S2 Dataset.**
(XLSX)

## Acknowledgments

Contributing members of the Dat'AIDS Study Group:

C. Chirouze, C. Drobacheff-Thiébaut, A. Foltzer, K. Bouiller, L. Hustache- Mathieu, Q. Lepiller, F. Bozon, O Babre, AS. Brunel, P. Muret, E. Chevalier (Besançon), C. Jacomet, H. Laurichesse, O. Lesens, M. Vidal, N. Mrozek, C. Aumeran, O. Baud, V. Corbin, E. Goncalvez, A Mirand, A brebion, C Henquell (Clermont-Ferrand), I. Lamaury, I. Fabre, E. Curlier, R. Ouissa, C. Herrmann-Storck, B. Tressieres, MC. Receveur, F. Boulard, C. Daniel, C. Clavel, PM. Roger, S. Markowicz, N. Chellum Rungen (Guadeloupe), D. Merrien, P. Perré, T. Guimard, O. Bollangier, S. Leautez, M. Morrier, L. Laine, D. Boucher, P. Point (La Roche sur Yon), L. Cotte, F. Ader, A. Becker, A. Boibieux, C. Brochier F, Brunel-Dalmas, O. Cannesson, P. Chiarello, C. Chidiac, S. Degroodt, T. Ferry, M. Godinot, J.M. Livrozet, D. Makhloufi, P. Miailhes, T. Perpoint, M. Perry, C. Pouderoux, S. Roux, C. Triffault-Fillit, F. Valour, C. Charre, V. Icard, J.C. Tardy, M.A. Trabaud (Lyon), I. Ravaux, A. Ménard, AY. Belkhir, P. Colson, C. Dhiver, A. Madrid, M. Martin-Degioanni, L. Meddeb, M. Mokhtari, A. Motte, A. Raoux, C. Toméi, H. Tissot-Dupont (Marseille IHU Méditerranée), I. Poizot-Martin, S. Brégigeon, O.

Zaegel-Faucher, V. Obry-Roguet, H Laroche, M. Orticoni, M.J. Soavi, E. Ressiot, M.J. Ducassou, I. Jaquet, S. Galie, H. Colson, A.S. Ritleng, A. Ivanova, C. Debreux, C. Lions, T Rojas-Rojas (Marseille Ste-Marguerite), A. Cabié, S. Abel, J. Bavay, B. Bigeard, O. Cabras, L. Cuzin, R. Dupin de Majoubert, L. Fagour, K. Guitteaud, A. Marquise, F. Najioullah, S. Pierre-François, J. Pasquier, P. Richard, K. Rome, JM Turmel, C. Varache (Martinique), N. Atoui, M. Bistoquet, E Delaporte, V. Le Moing, A. Makinson, N. Meftah, C. Merle de Boever, B. Montes, A. Montoya Ferrer, E. Tuaillon, J. Reynes (Montpellier), B. Lefèvre, E. Jeanmaire, S. Hénard, E. Frentiu, A. Charmillon, A. Legoff, N. Tissot, M. André, L. Boyer, MP. Bouillon, M. Delestan, F. Goehringer, S. Bevilacqua, C. Rabaud, T. May (Nancy), F. Raffi, C. Allavena, O. Aubry, E. Billaud, C. Biron, B. Bonnet, S. Bouchez, D. Boutoille, C. Brunet-Cartier, C. Deschanvres, B.J. Gaborit, A. Grégoire, M. Grégoire, O. Grossi, R. Guéry, T. Jovelin, M. Lefebvre, P. Le Turnier, R. Lecomte, P. Morineau, V. Reliquet, S. Sécher, M. Cavellec, E. Paredes, A. Soria, V. Ferré, E. André-Garnier, A. Rodallec (Nantes), P. Pugliese, S. Breaud, C. Ceppi, D. Chirio, E. Cua, P. Dellamonica, E. Demonchy, A. De Monte, J. Durant, C. Etienne, S. Ferrando, R. Garraffo, C. Michelangeli, V. Mondain, A. Naqvi, N. Oran, I. Perbost, M. Carles, C. Klotz, A. Maka, C. Pradier, B. Prouvost-Keller, K. Risso, V. Rio, E. Rosenthal, I. Touitou, S. Wehrlen-Pugliese, G. Zouzou (Nice), L. Hocqueloux, T. Prazuck, C. Gubavu, A. Sève, S. Giaché, V. Rzepecki, M. Colin, C. Boulard, G. Thomas (Orléans), A. Cheret, C. Goujard, Y. Quertainmont, E. Teicher, N. Lerolle, S. Jaureguiberry, R. Colarino, O. Deradji, A. Castro, A. Barrail-Tran (Paris APHP Bicètre), Y. Yazdanpanah, R. Landman, V. Joly, J. Ghosn, C. Rioux, S. Lariven, A. Gervais, FX. Lescure, S. Matheron, F. Louni, Z. Julia, S. Le GAC C. Charpentier, D. Descamps, G. Peytavin (Paris APHP Bichat), C. Duvivier, C. Aguilar, F. Alby-Laurent, K. Amazzough, G. Benabdelmoumen, P. Bossi, G. Cessot, C. Charlier, P.H. Consigny, K. Jidar, E. Lafont, F. Lanternier, J. Leporrier, O. Lortholary, C. Louisin, J. Lourenco, P. Parize, B. Pilmis, C. Rouzaud, F. Touam (Paris APHP Necker Pasteur), MA. Valantin, R. Tubiana, R. Agher, S. Seang, L. Schneider, R. PaLich, C. Blanc, C. Katlama (Paris APHP Pitié Salpétrière), F. Bani-Sadr, JL. Berger, Y. N'Guyen, D. Lambert, I. Kmiec, M. Hentzien, A. Brunet, J. Romaru, H. Marty, V. Brodard (Reims), C. Arvieux, P. Tattevin, M. Revest, F. Souala, M. Baldeyrou, S. Patrat-Delon, J.M. Chapplain, F. Benezit, M. Dupont, M. Poinot, A. Maillard, C. Pronier, F. Lemaitre, C. Morlat, M. Poisson-Vannier, T. Jovelin, JP. Sinteff (Rennes), A. Gagneux-Brunon, E. Botelho-Nevers, A. Frésard, V. Ronat, F. Lucht (St-Etienne), D. Rey, P. Fischer, M. Partisani, C. Cheneau, C. Mélounou, C. Bernard-Henry, E. de Mautort, S. Fafi-Kremer (Strasbourg), P. Delobel, M. Alvarez, N. Biezunski, A. Debard, C. Delpierre, G. Gaube, P. Lansalot, L. Lelièvre, M. Marcel, G. Martin-Blondel, M. Piffaut, L. Porte, K. Saune (Toulouse), O. Robineau, F. Ajana, E. Aïssi, I. Alcaraz, E. Alidjinou, V. Baclet, L. Bocket, A. Boucher, M. Digumber, T. Huleux, B. Lafon-Desmurs, A. Meybeck, M. Pradier, M. Tetart, P. Thill, N. Viget, M. Valette (Tourcoing).

C Chirouze: cchirouze@chu-besancon.fr

C Jacomet: cjacomet@chu-clermontferrand.fr

I Lamaury: isabelle.lamaury@chu-guadeloupe.fr

D Merrien: dominique.merrien@chd-vendee.fr

I Ravaux: isabelle.ravaux@ap-hm.fr

I Poizot-Martin: isabellepoizotmartin@gmail.com

J Reynes: j-reynes@chu-montpellier.fr

B Lefevre: b.lefevre@chru-nancy.fr

F Raffi: francois.raffi@chu-nantes.fr

L Hocqueloux: laurent.hocqueloux@chr-orleans.fr

A Cheret: antoine.cheret@aphp.fr

Y Yazdanpanah: yazdan.yazdanpanah@aphp.fr

C Katlama: christine.katlama@aphp.fr

F Bani-Sadr: fbanisadr@chu-reims.fr

C Arvieux: cedric.arvieux@chu-rennes.fr

A Gagneux-Brunon: amandine.gagneux-brunon@chu-st-etienne.fr

P Delobel: delobel.p@chu-toulouse.fr

O Robineau: olivier.robineau@ch-tourcoing.fr

## Author Contributions

**Conceptualization:** David Rey.

**Data curation:** Thomas Lemmet.

**Formal analysis:** Thomas Jovelin, Marine Maurel, Cyrille Delpierre.

**Methodology:** Thomas Lemmet, David Rey.

**Project administration:** David Rey.

**Supervision:** David Rey.

**Writing – original draft:** Thomas Lemmet.

**Writing – review & editing:** Thomas Lemmet, Laurent Cotte, Clotilde Allavena, Thomas Huleux, Claudine Duvivier, Hélène Laroche, André Cabie, Pascal Pugliese, Thomas Jovelin, David Rey.

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
