## [Decision Letter · Decision Letter 0]

2 Feb 2022

PONE-D-21-37034High syphilis prevalence and incidence in patients living with HIV and Preexposure Prophylaxis users: a retrospective review in the French Dat’AIDS cohortPLOS ONE

Dear Dr. Lemmet,

Thank you for submitting your manuscript to PLOS ONE. After careful consideration, we feel that it has merit but does not fully meet PLOS ONE’s publication criteria as it currently stands. Therefore, we invite you to submit a revised version of the manuscript that addresses the points raised during the review process.

We look forward to receiving your revised manuscript.

Kind regards,

Giuseppe Vittorio De Socio, MD, PhD

Academic Editor

PLOS ONE

Journal Requirements:

3. One of the noted authors is a group or consortium [Dat’AIDS Study Group]. In addition to naming the author group, please list the individual authors and affiliations within this group in the acknowledgments section of your manuscript. Please also indicate clearly a lead author for this group along with a contact email address.

Reviewers' comments:

Reviewer's Responses to Questions

**Comments to the Author**

1. Is the manuscript technically sound, and do the data support the conclusions?

Reviewer #1: Yes

Reviewer #2: Yes

2. Has the statistical analysis been performed appropriately and rigorously? 

Reviewer #1: Yes

Reviewer #2: Yes

3. Have the authors made all data underlying the findings in their manuscript fully available?

Reviewer #1: Yes

Reviewer #2: Yes

4. Is the manuscript presented in an intelligible fashion and written in standard English?

Reviewer #1: No

Reviewer #2: Yes

5. Review Comments to the Author

Reviewer #1: In their study, Lemmet et al. evaluated syphilis prevalence and incidence among patients living with HIV (PLWH) and

PrEP users in a large cohort in France, concluding that syphilis prevalence and incidence were high in these populations.

Here reported my comments:

1. The manuscript would benefit from thorough proofreading to better rephrase some sentences;

2. Please uniform numbers and acronym throughout the text;

3. Please modify PLWH from patients to people living with HIV;

4. Please add references for the sentences "In 2018 in France, 1762 new diagnoses of early syphilis have been notified, of which 79% occurred in MSM (“Santé publique France”). Although the number of cases seems to stabilize since 2016, the number of reported cases remains higher than in 2010, and 30 % are co-infected with HIV according to “Santé Publique France”.

5. A black/grey box is present in figure 2, 3 and 4 also when downloading the figures

6. There are some repetitions of data in the results section.

7. Do you have data on sex of sexual partner in PrEP cohort? Any statistical differences in characteristics of participants at enrollment between HIV and PrEP populations?

8. Could you please motivate the variables chosen in the multivariate models?

9. Could you enrich the part of discussion regarding to PrEP users?

10. In the discussion, you should reorganize the last part of the section (281-284 lines seem to anticipate any limitations)

11. Do you do any counseling in your cohort?

12. Did they receive penicillin or other treatments? Any neurosyphilis?

Reviewer #2: The article is interesting and well-written. Moreover, the statistical analysis il well-conducted. Nevertheless, a check of all the underlying assumptions of the multivariate logistic regression - independence of errors, linearity in the logit for continuous variables, absence of multicollinearity, and lack of strongly influential outliers - is missing. Such a check needs to be added in the revised version of the paper. Once the check is done, if the underlying assumptions do not hold for the available data, a nonparametric version of the considered method, making fewer assumptions, may be considered. Finally, I have the following minor comments:

Lines 134-135: briefly explain the aim of these tests: “Descriptive analysis” is too vague.

Lines 136-138: write the sentence better. Pay attention to singular and plural words.

Line 152 and Table 1: the symbol “n” is commonly used for the sample size. “N” is used, for the first time, in Table 1; this is unusual. Please clearly define the symbols “n” and “N”.

6. PLOS authors have the option to publish the peer review history of their article (what does this mean?). If published, this will include your full peer review and any attached files.

Reviewer #1: No

Reviewer #2: No

---

## [Author Response · Author response to Decision Letter 0]

31 Mar 2022

Please find attached to this new submission a rebuttal letter with response to the editor and the reviewers' requests.

---

## [Editor Report · Decision Letter 1]

4 May 2022

High syphilis prevalence and incidence in people living with HIV and Preexposure Prophylaxis users: a retrospective review in the French Dat’AIDS cohort

PONE-D-21-37034R1

Dear Dr. Lemmet,

We’re pleased to inform you that your manuscript has been judged scientifically suitable for publication and will be formally accepted for publication once it meets all outstanding technical requirements.

Kind regards,

Giuseppe Vittorio De Socio, MD, PhD

Academic Editor

PLOS ONE
---

## [Editor Report · Acceptance letter]

10 May 2022

PONE-D-21-37034R1 

High syphilis prevalence and incidence in people living with HIV and Preexposure Prophylaxis users: a retrospective review in the French Dat’AIDS cohort 

Dear Dr. Lemmet:

I'm pleased to inform you that your manuscript has been deemed suitable for publication in PLOS ONE. Congratulations! Your manuscript is now with our production department. 

Kind regards, 

on behalf of

Dr. Giuseppe Vittorio De Socio 

Academic Editor

PLOS ONE